# Recurrent Intraocular Lens Dislocation in a Patient with Familial Ectopia Lentis

**DOI:** 10.3390/ijerph18094545

**Published:** 2021-04-25

**Authors:** Tomasz K. Wilczyński, Alfred Niewiem, Rafał Leszczyński, Katarzyna Michalska-Małecka

**Affiliations:** 1University Clinical Center, University Hospital Medical University of Silesia, 40-514 Katowice, Poland; tomaszwil@wp.pl (T.K.W.); niewiem.alfred@gmail.com (A.N.); rafles3@wp.pl (R.L.); 2Department of Ophthalmology, School of Medicine in Katowice, Medical University of Silesia, 40-514 Katowice, Poland

**Keywords:** Carlevale, dislocation, familial ectopia lentis

## Abstract

A 36-year-old patient presented to the hospital with recurrent dislocation of the intraocular lens (IOL). The patient with the diagnosis of familial ectopia lentis was first operated on for crystalline lens subluxation in the left eye in 2007 and in the right eye in 2009. In both eyes, lens extraction with anterior vitrectomy and transscleral fixation of a rigid IOL was performed. In 2011, the IOL in the right eye luxated into the vitreous cavity due to ocular trauma. The patient underwent a pars plana vitrectomy with the IOL resuturation to the sclera. Seven years later, a spontaneous vision loss in the right eye was caused by a retinal detachment. The pars plana vitrectomy with silicone oil tamponade and a consequential oil removal three months later were performed in 2018. The follow-up examination revealed recurrent IOL dislocation in the same eye. Due to a history of previous suture-related complications a decision was made to remove the subluxated rigid polymethyl-methacrylate (PMMA) IOL and fixate to sclera a sutureless SOLEKO FIL SSF Carlevale lens. The purpose of this report is to present a single case of a 36-year-old patient who was presented to the hospital with recurrent dislocation of the intraocular lens. In a three-month follow-up period, a good anatomical and functional outcome was finally obtained with transscleral sutureless intraocular lens. This lens is an option worth considering especially in a young patient with a long life expectancy and physically active.

## 1. Introduction

Both natural and artificial intraocular lenses can become dislocated. Lens dislocation can be caused by multiple factors, i.e., trauma, genetic disorders including Marfan, Weill–Marchesani syndromes and familial ectopia lentis, or iatrogenic factors [1]. Familial ectopia lentis is characterized by bilateral, symmetrical subluxation of the lens in the superotemporal direction [2]. A disease trait is inherited in an autosomal dominant manner. The disease can occur at birth or develop later in life. In the case of a patient’s natural crystalline lens dislocation into the vitreous cavity, the treatment of choice is pars plana vitrectomy aimed to remove the dislocated lens [3]. Intraocular lens (IOL) implantation can be considered to restore the visual function of the affected eye and consequently bilateral vision [4]. A wide range of IOL implantation techniques that are presently available for an eye without capsular support along with the latest generation of intraocular lenses makes the implantation procedure faster and less traumatic [5]. The basic complications of transscleral IOL fixation include (1) suture exposure that can lead to exacerbation of local inflammatory response and (2) mechanical damage to sutures with possible lens decentration. Moreover, suture fixating the IOL to the scleral wall can degrade over time, which is of particular concern in younger patients with high life expectancy [6]. Fewer postoperative complications have been observed after sutureless IOL fixation [7].

The newest developments in lens implants are the iris claw lenses, which can be fixed anteriorly to the iris or in the posterior chamber. At present, fewer models of iris-claw lenses are available on the market, among others, the Artisan and Verisyse lenses. Visual rehabilitation is shorter compared to scleral fixated implants, there are fewer postoperative complications, and visual acuity stabilizes faster.

At this point, SOLEKO FIL SSF Carlevale lens should definitely be mentioned. This single-piece acrylic foldable intraocular lens has special anchor plugs that facilitate sutureless fixation to the scleral wall [Figure 1]. Operative times are shortened and the risk of injury to neighboring tissues is reduced. The implantation technique involves making two punctures with a 23-G needle below the scleral flaps 1.5–2 mm from the limbus, 180° apart. Anchor plugs are then driven through the incision. After that, they reopen and fixate onto the sclera. Scleral flaps are then secured with sutures.

## 2. Methods—Case Report

A 36-year-old patient had been ophthalmologically observed since early childhood due to congenital bilateral superotemporal subluxation of the lens. The ophthalmologist considered Marfan syndrome in the differential diagnosis, but no abnormalities have so far been found regarding any other organ systems. The patient gave birth to two children. The younger son has been diagnosed with bilateral lens ectopia, while the older son does not exhibit any ocular abnormalities. The patient refused to give consent to genetic testing, as well as for the children.

Bilateral lens ectopia was monitored at the Ophthalmology Department. The refractive error—myopia—was regularly corrected ultimately stabilizing at −5.0 diopters. In September 2007, due to severe left eye visual acuity decrease, a decision was made to operate the patient for a subluxated lens. Extraction of the subluxated lens was performed along with anterior vitrectomy and trans-scleral fixation of the intraocular lens. Visual acuity in the left eye improved from 1/50 with −5.0 diopters correction to 5/25 with no correction. Two years later, the procedure was performed on the right eye with visual acuity improvement from 1/50 to 5/25 with no correction. The patient was followed up for the next two years; no postoperative complications were seen. Due to ocular trauma in 2011, the intraocular lens luxated into the vitreous cavity in the right eye. Pars plana vitrectomy was performed with the IOL resutured into the sclera. The patient was discharged with uncorrected visual acuity of 5/50. For the next six years, there was kept regular follow-up appointments without reporting any complaints. In August 2018, the patient was admitted to the Ophthalmology Department with a progressive visual acuity decreased in the right eye. Retinal detachment with macular involvement was diagnosed while both IOLs were correctly positioned. The patient underwent pars plana vitrectomy with silicone oil endotamponade and successful retinal reattachment. Three months later, the silicone oil was removed. A follow-up examination revealed right IOL dislocation. A decision was made to extract the subluxated rigid polymethyl methacrylate (PMMA) lens (fixated at 3th and 9th hour); a wide 5-mm incision was made at the limbus (to remove rigid polymethyl methacrylate (PMMA) intraocular lens.) A sutureless SOLEKO FIL SSF Carlevale lens was fixated at 12th and 6th hour. Considering several previous interventions (suture scleral fixation) and the risk of another dislocation, a sutureless option was selected.

## 3. Results

No postoperative complications were found; the posterior chamber IOL was well-positioned and centered. Three months after surgery, the best-corrected visual acuity in the right eye was 5/25, intraocular pressure was 16 mm Hg, the corneal incision was well-sealed, and corneal sutures were in place. There were no signs of IOL dislocation, and the retina remained reattached. Right eye keratometry revealed high irregular astigmatism—a result of corneal sutures placed on the wide incision made to extract the PMMA lens.

## 4. Discussion

In patients with natural crystalline lens dislocation, the choice between spectacle correction and surgical intervention depends on the degree of lens displacement and opacity, both of which affect visual acuity. Another consideration is glaucoma secondary to subluxation of the crystalline lens. The standard surgical technique developed to manage congenital lens ectopia is lensectomy with anterior vitrectomy. Literature reports numerous techniques of intraocular lens implantation in patients with no capsular support. Adequate stability can be achieved with anterior chamber IOLs, iris-claw lenses fixated on the anterior or posterior iris surface, and intrascleral fixated IOLs. Our patient had undergone two surgical procedures aimed at scleral IOL fixation that failed to succeed. It was therefore decided to use sutureless fixation, which, according to the literature, is more effective in securing the IOL in the right position [5]. The complication of the above methods may include retinal detachment, iritis, and irregular pupil [8].

In summary, patients with familial ectopia lentis should undergo regular ophthalmic examinations. Treatment requires careful analysis of the patient’s clinical condition and the risk of ophthalmic complications associated with natural crystalline lens extraction. IOL type and implantation modality should be selected based on the patient’s lifestyle, life expectancy, concomitant eye disease, and eyeball anatomy. In our opinion, a transscleral sutureless intraocular lens is an option worth considering, especially in a young patient with a long life expectancy and physically active. In addition, guided by the choice of method, the goal was to minimize possible intra- and post-surgical complications. Rossi et al. suggest good results of treatment with transscleral sutureless intraocular lenses with a few complications [9]. In our experiments, this lens is less traumatizing for eye tissues.

## 5. Key Points

Familial ectopia lentis is a rare disease with a lot of surgery-related complications;A regular follow-up is crucial for maintaining patients’ satisfactory visual acuity;A transscleral sutureless intraocular lens is an option worth considering, especially in a young patient with a long life expectancy and physically active [Figure 2 and Figure 3].

## Figures and Tables

**Figure 1 ijerph-18-04545-f001:**
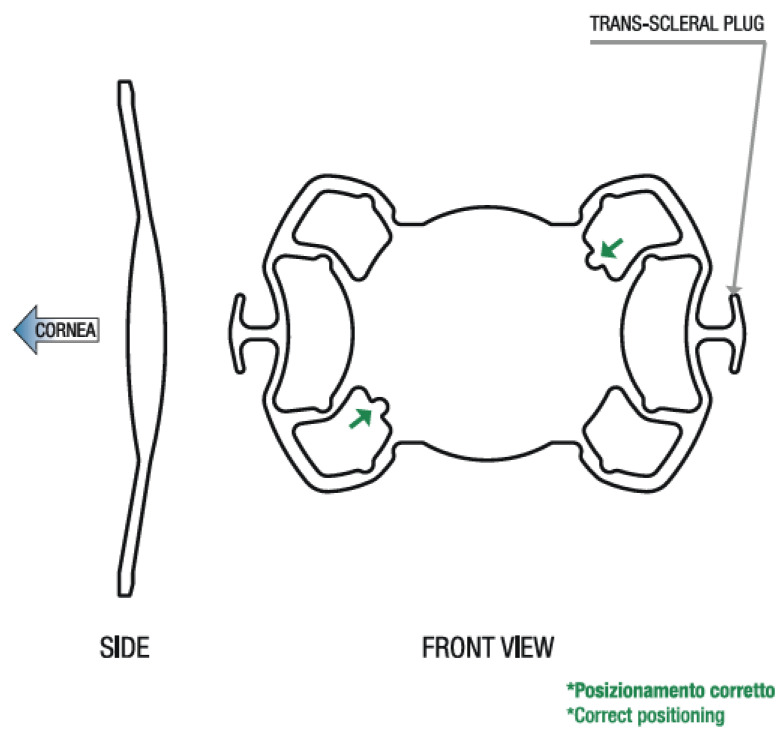
SOLEKO FIL SSF Carlevale lens.

**Figure 2 ijerph-18-04545-f002:**
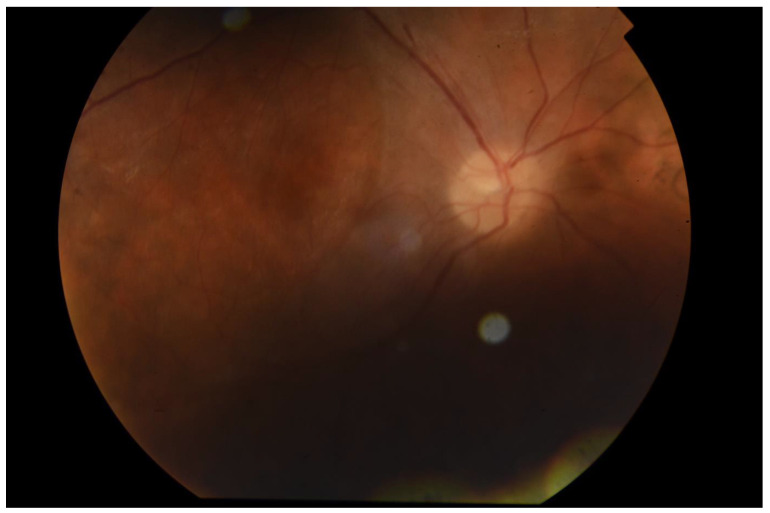
Right eye—posterior segment at 1 month of SSF intraocular lens (IOL) implantation.

**Figure 3 ijerph-18-04545-f003:**
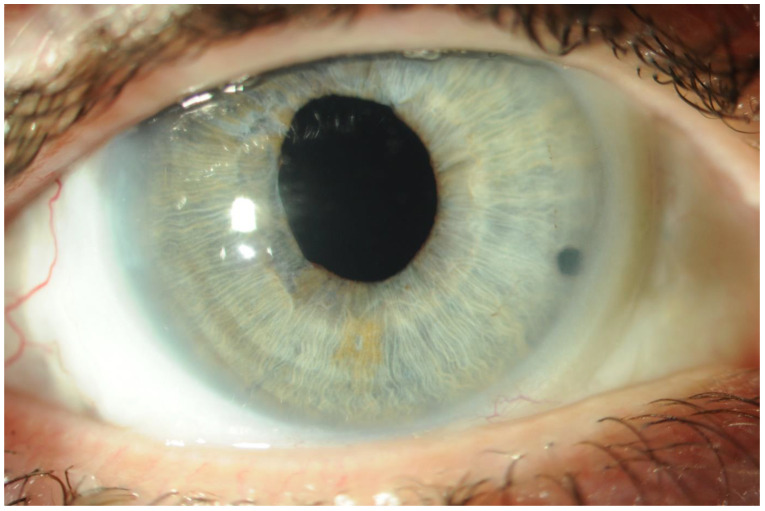
Right eye—anterior segment at 1 month of SSF IOL implantation.

## Data Availability

Data is contained within the article or supplementary material.

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
