# Peer review of "Recurrent Intraocular Lens Dislocation in a Patient with Familial Ectopia Lentis"

_ijerph, 2021, doi:10.3390/ijerph18094545_

Round 1

Reviewer 1 Report

Very patient- centred report.. good.

Please list complications which occur with other methods but did not occur in this case.

Limitations of the study

Increase number of references particularly those with a larger number of cases series. and are more recent 

Declare any/no conflict of interest.

Author Response

Thank you for your review.
I added a complication list of other methods and updated the reference list. At the end of the case report, I put a declaration of no conflict.

Reviewer 2 Report

This manuscript is well organised and written. Still, the 3-month follow-up period can be mentioned in the results section of the abstract. This paper requires minor English language spell checks and is recommended for publication after that. 

I would recommend to authors to add these two papers
in the introduction of their manuscript: 1) http://dx.doi.org/10.1136/bjophthalmol-2012-302921
(https://bjo.bmj.com/content/97/7/942.abstract) 2) https://www.ncbi.nlm.nih.gov/pmc/articles/PMC3930276/

Author Response

Thank you for your review and the submitted publications. During the literature review, I did not notice them. I used one of them and attached it to my references.

Reviewer 3 Report

During the entire treatment and follow-up of this case, the advantages of the sutureless fixation IOL given in the conclusion did not seem to be reflected.

According to the article, the patient's left eye remained stable after surgery in 2007, while two IOL luxate in the right eye seemed to be related to ocular trauma and retinal detachment. Despite the factors for retinal detachment, the second dislocation of the intraocular lens in the right eye occurred in the seventh year after surgery. During this period, according to the article, there are no complications and postoperative visual acuity is stable. However, the surgery about a sutureless Soleko Fil SSF Caravale Lens in the right eye has only been done for more than two years, which cannot reflect its advantages such as not easy to dislocate. And there are few postoperative follow-up data in the article, only one follow-up data for three months after surgery.

Moreover, the recording of visual acuity in the article is relatively disordered, and uncorrected visual acuity and corrected visual acuity appear alternately, which may be caused by the lack of data collection. However, this will lead to some difficulties in comparing the efficacy of surgery and lack of a unified vision acuity recording standard.

And the article does not reflect the follow-up record of the state of the patient's left eye, because in the end, the fixation of the intraocular lens in the left and right eyes is different, perhaps more conclusions can be drawn from this case by comparing the left and right eyes.

In short, I think the patient needs to be followed up for a longer period of time in order to reflect the conclusion that the article wants to express.

Author Response

Thank you for your review.
Our work was aimed at showing the consideration we encountered when making decisions about treating a patient. The more that the patient suffers from a rare disease that causes more complications during and after the procedure. By comparing two different methods of treatment for a patient with this disease, it is possible to see which method is more effective in the long term. It follows that standard methods of lens fixation can be very effective, but in more difficult cases other methods should be considered, more effective in the light of the literature. We know that the 3-month observation period is short, but we are constantly extending the observation period, and we plan to extend the study sample to more cases.

Round 2

Reviewer 3 Report

If the purpose of this case is not to compare the advantages and disadvantages of the two intraocular lens fixation methods, but to show your considerations and decisions when deciding to treat patients, then there is no problem. I also expect you to follow up the patient for a longer time to get deeper results.